# Antitumor Effects of Selenium

**DOI:** 10.3390/ijms222111844

**Published:** 2021-10-31

**Authors:** Seung Jo Kim, Min Chul Choi, Jong Min Park, An Sik Chung

**Affiliations:** 1Sangkyungwon Integrate Medical Caner Hospital, Yeoju 12616, Gyeonggido, Korea; sjkim@lntemed.kr; 2Comprehensive Gynecological Cancer Center, CHA Bundang Medical Center, Seongnam 13497, Gyeonggido, Korea; oursk79@cha.ac.kr; 3Oriental Medicine, Daejeon University, Daejeon 34520, Korea; jmpark@dju.kr; 4Department of Biological Sciences, Korea Advanced Institute of Science and technology, Daejeon 34141, Korea

**Keywords:** selenium compounds, ROS, apoptosis, metastasis, treatment of advanced cancer patients

## Abstract

Functions of selenium are diverse as antioxidant, anti-inflammation, increased immunity, reduced cancer incidence, blocking tumor invasion and metastasis, and further clinical application as treatment with radiation and chemotherapy. These functions of selenium are mostly related to oxidation and reduction mechanisms of selenium metabolites. Hydrogen selenide from selenite, and methylselenol (MSeH) from Se-methylselenocyteine (MSeC) and methylseleninicacid (MSeA) are the most reactive metabolites produced reactive oxygen species (ROS); furthermore, these metabolites may involve in oxidizing sulfhydryl groups, including glutathione. Selenite also reacted with glutathione and produces hydrogen selenide via selenodiglutathione (SeDG), which induces cytotoxicity as cell apoptosis, ROS production, DNA damage, and adenosine-methionine methylation in the cellular nucleus. However, a more pronounced effect was shown in the subsequent treatment of sodium selenite with chemotherapy and radiation therapy. High doses of sodium selenite were effective to increase radiation therapy and chemotherapy, and further to reduce radiation side effects and drug resistance. In our study, advanced cancer patients can tolerate until 5000 μg of sodium selenite in combination with radiation and chemotherapy since the half-life of sodium selenite may be relatively short, and, further, selenium may accumulates more in cancer cells than that of normal cells, which may be toxic to the cancer cells. Further clinical studies of high amount sodium selenite are required to treat advanced cancer patients.

## 1. Introduction

Selenium (Se) is an essential trace element for animals and human, and it has several important functions, such as cancer, antioxidant, increasing immunity, and antiviral activity. Se compounds have emerged as nutritional supplements with the most consistent anticancer effects among a number of micronutrients treated in animal experiments and further clinical trials in human [1]. Se deficiency is related to the occurrence of chronic disease, including certain tumor in human [2,3,4]. Cancer is a leading cause of death for people in the world. In 2020, more than 19 million new incidences occurred, and nearly 10 million cancer death were account for, estimated by WHO. Most of cancer death is mainly related to tumor metastasis.

A large body of evidence indicates that Se is physiologically important to prevent cancer incidence and block tumor metastasis. Se was first mentioned as possibly protective against cancer risk almost 60 years ago [1]. Epidemiological studies have been shown that a population with low Se intake and low plasma Se levels have an increased incidence of cancer, including cancer of breast, lung, stomach, bladder, ovaries, pancreas, thyroid, esophagus, head and neck, cerebellum, and melanoma [5,6,7,8,9,10,11,12]. Most of these studies have found Se status to be inversely associated with cancer risk. We would like to emphasize here that higher amount of sodium selenite (super nutritional levels) could block cancer incidence, tumor invasion, and metastasis, and, further, high-level of sodium selenite can be used to adjuvant therapy of radiation and anticancer drugs in patients with advanced cancer [13,14]. Our clinical treatment to advanced cancer or terminal cancer patients are tolerable to 5 mg or more doses of sodium selenite. In the clinical trials, a high dose of Se returns to almost a normal level in serum less than 260 μg/mL 24 h after injection. We postulate that relatively short half-life of Se and more absorption of sodium selenite in cancer cells would be more become toxic to cancer cells and therefore pose less toxicity to normal cells. It seems to be sodium selenite an excellent drug to treat cancer patients. However, further studies are required to define the most therapeutic doses of sodium selenite in each cancer patients in situation of anticancer drugs, radiation, drug resistance, stages of cancer, and, further, side effects of the treatment.

As mentioned, in organic Se compounds, such as MSeC, its metabolite, methylselenol, can generate radicals to induce apoptosis and blocking tumor invasion. Sodium selenite is also oxidants as accepting electrons without involving oxygen atoms. A simplified mechanism is disulfide isomerase (PDI), which can convert to inactive disulfide form according to following formula, as shown in the prevention of corona virus infections, which was hypothesized by Kieliszek and Lipinski (PDI-(SH)^2^ + Se^4+^ -> PDI-SS-PDI + Se^2+^) [15]. The hypothesis can apply to sodium selenite and MSeC, which have been implicated as an important factor in aging, antiviral activities, and aging related diseases, especially cancer, and, further, subsequent treatment of high doses of sodium selenite with radiation and chemotherapy has been performed to advanced cancer patients at several Institutions.

## 2. Metabolism of Selenium Compounds

Se exists in both organic and inorgarnic forms in natural diets. Organic Se is presented in the form of selenometathionine (SeM), selenocyteine, and MSeC, whereas inorganic forms occurs either as selenite or selenate. Among organic forms, SeM is the predominant forms of Se in Se-enriched yeast [16]. Se-methylselenocysteine (MSeC) is most abundant in garlic, broccoli, walnut, and some other plant products. Selenite, an inorganic form, has been used for laboratory and clinical trials, even though it has some toxicity. The various degree antitumor activity of different chemical forms of Se may be related to their metabolism in vivo. A flowchart describing the events in Se metabolism is presented in Figure 1. Se is important due to its localization within the active site of some selenoproteins with several functions and, further, the maintenance of redox balance in cells. Most Se compounds are readily absorbed from the diet and are mainly metabolized in the liver. Hydrogen selenide from selenite and MSeH from MSeC are important metabolites related to antitumor activities. Selenomethionine gets incorporated into the general body proteins as methionine. Therefore, it is less effective compared to MSeC and selenite. Metabolism of selenite is tightly regulated and is reduced to selenide. This selenide and SeDG serve as a precursor for the synthesis of selenoproteins, such as GPx and TrxR. Moreover, Se can be methylated mono-, di-, and tri-methyl forms of Se. This methylation is reversible in vivo. However, MSeC is directly converted to MSeH by β-lyase. In our studies, MSeC is the most effective Se compound to induce apoptosis, as well as block tumor invasion and metastasis, in cancer cells [17,18,19]. More studies on MSeC in vivo are required to measure pharmacokinetic of this compound and to determine half-life and maximum tolerable doses of MSeC. This compound is commercially available now. Even further, it would be interesting to see clinical application of this compound in cancer patients.

The ROS production by Se is shown in Figure 2. Hydrogen selenide from selenite and MSeH from MSeC react with GSH or sulfhydryl groups and produce O_2_^−^ and, further, H_2_O_2_. Selenide produces reactive oxygen species (ROS), such as superoxide anion and H_2_O_2_, which induce DNA single-strand breaks, the cell cycle arrest by blocking S/G2 phase and, further, cell death by apoptosis [20,21,22]. Selenide induced apoptosis through this mechanism as the genotoxic and pro-apoptotic effects by the production of superoxide anion on leukemia. Mammary or prostate cancer cells have been shown to be blocked by a superoxide dismutase or its mimics [23] but not by a hydroxyl radical scavenger [24]. Further, catalase added to the cell culture medium blocked the induction of cell death by selenite [25]. Selenite is metabolized to hydrogen selenide and induces DNA single-strand breaks and, subsequently, cell death by a combination of acute lysis (necrosis) and apoptosis in several cell lines [24,26,27,28]. Selenite-induced cell death is independent of these death proteases [17,29,30,31].

On the other hand, the anti-carcinogenic effects of Se mediated byCH_3_SeH or its derivatives have been shown in the papers by References [32,33,34,35,36,37,38]. They have shown that the CH_3_SeH-precusor MSeCN and MSeC can inhibit mammary cell growth, arresting cells in G1 phase and inducing apoptosis [21,22,30,39,40,41,42]. This effect is related to caspase-dependent apoptosis [31]. Methylselenol induced apoptosis by ROS production, subsequently altered mitochondrial membrane potential, and, further, induced caspases’ activity. The production of ROS triggers cytochrome *c* release into cytoplasm, which can activate either caspase-9 or -8 and, further, caspase-3, and, subsequently, induce apoptosis. In addition, MSeH induced apoptosis through the direct oxidation of vicinal sulfhydryl groups within the catalytic domains of cellular enzymes (e.g., protein kinase C) and generation of superoxide anion and other ROS by MSeH [43]. The difference in the cytotoxicity between selenide from selenite and MSeH from MSeC was well discussed in a recent review paper of Reference [44]. Selenide is very reactive and may redox cycle with oxygen, form elemental Se, and undergo methylation to monomethylselenol, dimethylselenide or trimethylselenium or incorperated into selenoproteins as selenocysteine. The methylation procedure is reversible, and there are methylation and demethylation reactions by methytransferases and demethyltransferases, respectively [45]. The most predominant excretory urinary metabolites are seleno-methyl-*N*-acetylgalatosamine, seleno-methyl-*N*-acetylglucosamine, and trimethylselenium [46,47].

## 3. Effects of Selenium on Cancer Cells Depended on Its Concentration and Species

The viability of human breast cancer cells was inhibited in vitro in a dose-dependent manner. By Se supplementation, however, a normal cell line was relatively resistant to Se concentration [48,49]. It has been demonstrated that neoplastic breast tissues contain a high amount of Se compared to its surrounding non-neoplastic tissues [50]. Similar results have been reported in colorectal cancer and gastric cancer patients by the same group. In conjunction, several investigations clearly indicated increased expression of the selenoprotein, TrxR in the neoplastic tissues compared to the normal tissues [51]. Neoplastic transformation is often associated with increased cell proliferation during initiation and promotion of tumor development. These observations, together, imply the importance of Se in the established neoplastic tissue. On the other hand, a high dose of Se compounds inhibits neoplastic growth by production ROS [52]. Redox active Se compounds are evolving as promising chemotherapeutic agents through tumor selectivity and multi-target response, which are of great benefit in preventing development of drug resistance. Generation of ROS is implicated in Se-mediated cytotoxic effects on cancer cells. Recent findings have indicated that activation of diverse intracellular signaling leads to cell death in single cell line upon treatment with Se compounds, including selenite and SeDG. Both selenite and SeDG exhibited similar toxicity but different mechanisms. Morphologically and molecular SeDG induced apoptosis-like cell death. The above results imply that the diverse cytotoxic effects and variable potential redox active Se compounds on the survival of the cells and, thereby, substantiate the potential of chemical species-specific usage of Se in the treatment of cancers.

Generation of ROS from the reaction of Se compounds with thiols was shown in mammary tumor cells [53]. It was suggested that a free radical, the superoxide anion (O_2_^−^), and H_2_O_2_ are produced from the reaction of selenite and cysteine as glutathione. The capacity of cells to form Se metabolites that more easily oxidize glutathione and other thiols in the tumor cells produce ROS and peroxides. A significant difference in thiol concentration between cancer cells and normal cells has been observed [54]. Treatment with high amount of sodium selenite selectively eliminates cancer cells by generating more free radicals than that of normal cells without a systemic Se toxicity [55]. The mechanism of selenite penetrates in the cell is shown in Figure 3 in this paper. Selenite coverts hydrogen selenide by cysteine oxidation in extracellular compartment. Both hydrogen selenide and cystine get into the intracellular compartment by the χ_c_^−^ cystine/glutamate antiport, which reduces intracellularly to cysteine by NADPH-dependent redox systems. A significant fraction of intracellular cysteine is resected back to the extracellular compartment by multidrug resistant protein transport systems. More hydrogen selenide can be absorbed in the cancer cells, which is the most reactive Se compound [56]. Metastatic tumor or drug-resistance tumor cells absorb more hydrogen selenide, which can be more toxic to tumor cells by the glutathione oxidation-reduction system. Glutathione is the most stable form of cysteine in the cells and, further, can be synthesized more under oxidative stress, such as cancer cells.

## 4. Selenium on Immune Functions and Anti-Inflammatory Activity

Similar to other cell types, immune cells respond to amount of dietary Se by increasing expression of many selenoproteins, although not all selenoproteins are equivalently affected. GPx1 and GPx4 in human lymphocytes were increased by supplementation with sodium selenite (50 μg or 100 μg/day) [57], similar to mice studies [58]. Se activates immune functions via the activation of IL-2 receptor [59]. IL-2 activated by T helper cells increases the activity of immune system and differentiates immune cells. The antioxidant functions of Se by the action of selenoproteins can scavenger ROS and protect cells, and Se activates the IL-2 receptor, which can activate the immune cells and regulate the immune functions.

Se has both antioxidant and anti-inflammatory functions. These functions may be explained by the roles of selenoproteins, such as GPx, TrxR, selenoprotein P, and selenoprotein W [35,60,61,62]. The roles of these enzymes have been extensively reviewed in relation to their potential functions in physiological phenomena as antioxidant defense, redox regulation of cytokine, differentiation, apoptosis, and tumorigenesis [63]. Glutathione-peroxidase and other selenoproteins can reduce hydrogen peroxide and phospholipid peroxides, thereby blocking the propagation of free radicals and reactive oxygen species, and they can also reduce hydrogen peroxides intermediate in the cyclooxygenase and lipoxygenase pathways diminishing the production of inflammatory prostaglandin and leukotrienes [64]. Any condition associated with increased oxidative stress or inflammation might be expected to be influenced by Se levels, which may be the case in rheumatoid arthritis, pancreatitis, and asthma. Supplementation with 200 μg Se in a group of rheumatoid arthritis patients for three months significantly reduced pain and joint involvement [65]. Administration of 600 μg Se along with other antioxidants to patients with chronic and recurrent pancreatitis significantly reduced pain and frequency attacks [66]. A protective relationship was found between dietary Se intake and asthma in a large occupation-based case control study [67].

Se deficiency could increase cancer risk, which might be expected on the basis of the known functions of selenoproteins in antioxidant protection and redox regulation and, thus, in the metabolic defense against carcinogenic free radicals. Mutagenic oxidative stress is generally thought to be a major factor in the initiation of human carcinogenesis, as the electron-rich DNA bases are susceptible to electrophilic attack by reactive oxygen species (ROS), including superoxide radical, hydrogen peroxide, hydroxyl radicals, and electrophilic metabolite of xenobiotics and other reactive intermediated metabolites. These ROS can cause genetic damage and the reproduction of mutant oncogenes and tumor suppressor genes, as well as epigenetic changes that affect expression. Se protection against such changes has been shown in the results from the induction of skin tumors by other ultraviolet irradiation [68,69] or the levels of phorbol esters [70], which was inversely related to skin GPx activity in animal models, and protection by selenite against (2-oxopropyl) amine-induced intrahepatic cholangio carcinomas in Syrian golden hamsters, which was correlated with the restoration of hepatic GPx activity [71].

Se plays an important role in defense against acute illness, such as sepsis syndrome. Se concentration was decreased during an intensive care unit (ICU) stay in all groups, as well as severity of sepsis syndrome, such as systemic inflammatory response and multiorgan failure. Plasma Se was inversely related to admission acute physiology [71], and treatment of high-dose sodium selenite reduced the mortality rate in patients with severe sepsis and, especially, in septic shock [72]; high doses of Se also reduced ventilator-associated pneumonia and illness severity in critically ill patients with systemic inflammation [73]. Mechanisms of the beneficial effects of Se treatment on sepsis were postulated by blocking lipopolysaccharide production, as well as inhibition of myocardial cytochrome c oxidase activity and mechanically ventilated septic patients. Recently, intravenous administration of high doses Se did not improve outcome results from severe sepsis patients guided by procalcitonin therapy algorithm [74]. There may be a few debates against using high-doses of Se to sepsis. However, there was strong evidence that Se might enhance the activities of important selenoenzymes involved in the maintenance of redox-homeostasis, as well as immune and endothelial functions.

## 5. Blocking Tumor Invasion and Metastasis

Metastasis is one of the major causes of cancer mortality. Earlier studies have shown that dietary supplementation of Se reduces lung metastasis of melanoma cells, and Se-enriched yeast inhibits the spread of Lewis lung carcinoma cells in mice [73,74,75]. These findings collectively suggest that a nutritional adjuvant containing Se may be beneficial in reducing metastasis.

Matrix metalloproteinases (MMPs) play a major role in promoting angiogenesis and tumor metastasis [76]. MMP-2 and MMP-9 are crucial enzymes in the process of tumor metastasis derived from ECM degradation [76]. Several experiments by our group demonstrate that selenite inhibits tumor invasion by blocking MMP-2 and -9 expression [28], and MSeA inhibits tumor invasion induced by PMA via blocking MMP-2 activation [77]. Another experiment indicates that MSeH reduces metastasis of melanoma cells as a nutritional adjuvant due to suppression of integrin expression and the inhibition of MMPs [78].

Another experiment has shown that MSeA, in contrast to selenite, specifically induced human macro-vascular endothelial (HUVEC) G1 cell cycle arrest [42] and apoptotic death by caspase-mediated execution [31], inhibited endothelial MMP-2 expression [79,80], and inhibited cancer epithelial expression of VEGF [80]. Regarding its anti-proliferative action, it has been reported that MSeA can inhibit HUVEC cell cycle G1 to S progression stimulated by angiogenic factors [42], but the mechanism remains to be elucidated.

Finally, Se blocks tumor invasion and metastasis by inhibition of MMPs. These MMPs are involved in ECM degradation, basement membrane invasion, intravasation, extravasation, and, further, angiogenesis. Furthermore, animal and human intervention studies are required to establish whether Se compounds can be effectively used for chemoprevention and blocking tumor metastasis.

## 6. Clinical Application of High Doses of Sodium Selenite with Radiation Therapy and Chemotherapies for Advanced Cancer Patients

Selenite by itself is efficient in targeting certain cancers. However, a more pronounced effect has been shown in the subsequent chemotherapy and radiation therapy. The metastatic cancer patients were orally administrated by selenite (5.5 to 49.5 mg) as single dose 2 h before each radiation therapy. It was shown that sodium selenite was effective to increase radiation therapy, reduce radiation side effects, such as pain, stabilize the cancer status, and, further, to reduce PSA in prostate cancer patients [81]. It was measured that the half-life of sodium selenite was 18.5 h, and no adverse effects were shown at the level of selenite until 33 mg dose. The results from a phase 1 study of sodium selenite were safe and efficient to treat patients with metastatic cancer in combination with palliative radiation therapy until a 33 mg dose. Similar results were shown in other study [82]. In this study, 34 patients with different chemo-resistant tumors received intravenously sodium selenite daily for five consecutive days, either two weeks or four weeks. Plasma half-life of Se was 18.25 h, and the maximum tolerated dose was 10.2 mg/m^2^ in cancer patients with chemotherapy. The most common adverse events were fatigue, nausea, and cramps in fingers and legs. Dose limiting toxicities were acute of short duration and reversible. Biomarkers for organ functions indicated no major systemic toxicity. It was concluded that sodium selenite is safe and tolerable when administrated under their current protocol. It would be interesting to see that the result of ongoing prolonged infusion study with sodium selenite is more effective treatment for advanced cancer patients. Another phase 1 trial of sodium selenite plus chemotherapy was shown in gynecological cancer patients [83]. Sodium selenite was administrated to the patients from 50 μg to 500 μg doses. Grade 3/4 toxicities included neutropenia, pain, infection, neurologic, and pulmonary adverse effects. The maximum tolerated dose was not reached. Se had no effect on carboplatin pharmacokinetics in the treatment of chemo-naive women with gynecologic cancers. Correlative studies showed post-treatment downregulation of RAD51AP1, a protein involved DNA repair, in both cancer cell lines and patient tumors. The addition of Se to carboplatin/paclitaxel treatment was safe and well tolerated, and it did not alter carboplatin pharmacokinetics. A 5000 μg dose of sodium selenite is suggested as the dose of evaluated in a phase II clinical trial in their ongoing studies. In our study, several terminal cancer patients could tolerate more than 5000 μg of sodium selenite with chemotherapies and radiotherapy. Serum Se concentrations were monitored every time at 24 h after injection of high doses of sodium selenite and was reduced to less than 260 μg/mL, even more than 5000 μg of sodium selenite treatment at Sangkyungwon Integrate Medical Cancer Hospital (SIMCH) (unpublished data; Table 1) in South Korea. Three hundred and ten patients were supplemented by high dose of sodium selenite with advanced drug-resistant metastatic cancers, such as breast, gynecological, gastrointestinal tract, liver cancer, pancreatic, gall bladder, renal, bladder cancers, and lymphoma, from July 2019 to August 2021. The results and procedures of this clinical trial are shown here (unpublished data). During and after a high dose of the selenite treatment, no irreversible adverse events were found in most cancer patients, but a few reversible mild gastroenteric symptoms were found. After the injection of the high dose of sodium selenite, Se levels were less than 260 μg/mL at 24 h, which may not be toxic. We present two advanced gynecologic cancer patients successfully treated with high doses of sodium selenite in which continual treatment reduced the size of tumors and improved other symptoms, as shown in Figure 4. 

It has been postulated that a short half-life of sodium selenite and more accumulation of the Se in the cancer cells may be more toxic in cancer cells than that in normal cells. As the patients were treated with multiple lines of chemotherapy, they eventually showed drug resistance. The patients were treated with several drugs and showed drug resistance. High doses of sodium selenite were applied to these chemo-resistant patients, and some of them were terminal patients. The detailed segment modification of dosage and duration were performed upon individual cases for 3 months, 6 months, and 12 months and reevaluated, including quality of life. The results suggest that high doses of sodium selenite as conventional treatment with chemotherapy and radiation can be considered a prominent method. An integrated medical treatment of sodium selenite with chemotherapy, radiation, immunotherapy, surgery, and resistance to chemotherapy may be applied to advanced cancer patients. In the future, it would be interesting to find that each cancer patient can apply modification doses and duration in treatment with or without combination of standard procedure, such as surgery, irradiation, chemotherapy, and, further, immunotherapy.

In conclusion, high dose Se treatment can be a possible therapy in some cases for advanced cancer patients. We discuss the pertinent literature in a mine review style and also describe, through our recent clinical understanding, that Se may be tolerable agent without any significant severe adverse events as an integrative medical treatment combination with chemotherapy, radiotherapy, and radical surgery in the advanced stage cancer of patients in chemo-resistant status or to prevent developing it. In the beginning, the individual takes a modification in therapy of dose of duration of treatment with or without combination of standard treatment, such as surgery, irradiation, chemotherapy, and also immunotherapy. In the future, randomized controlled prospective clinical trials in terms of high dose inorganic Se treatment will be necessary to be carried out at various institutions internationally.

## Figures and Tables

**Figure 1 ijms-22-11844-f001:**
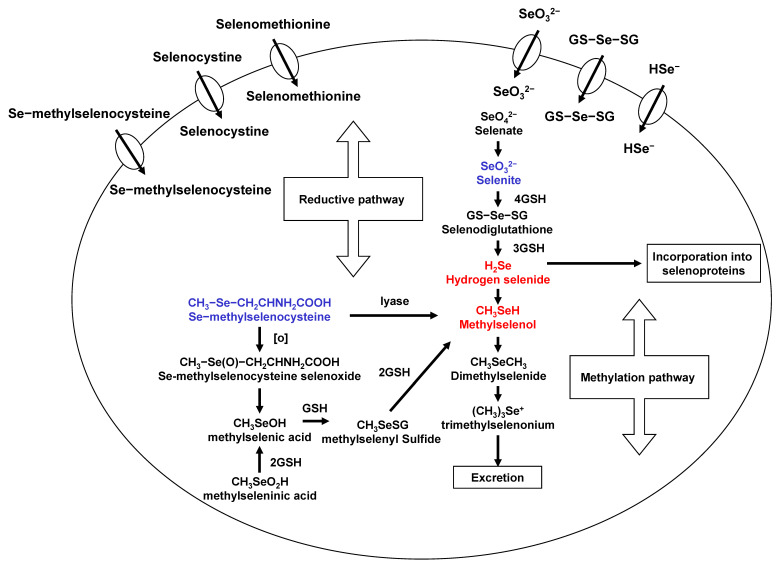
Selenium metabolism.

**Figure 2 ijms-22-11844-f002:**
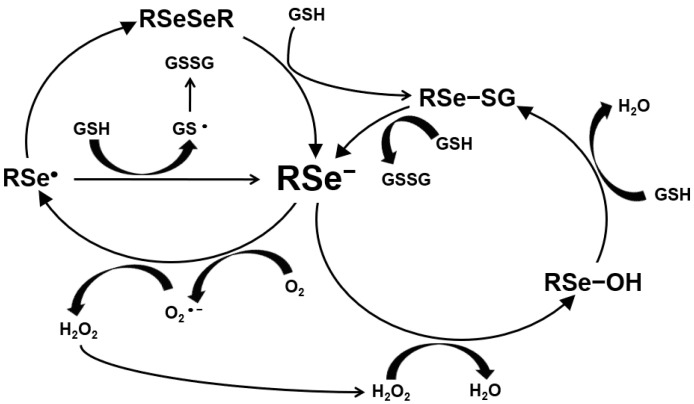
Production of ROS by selenium.

**Figure 3 ijms-22-11844-f003:**
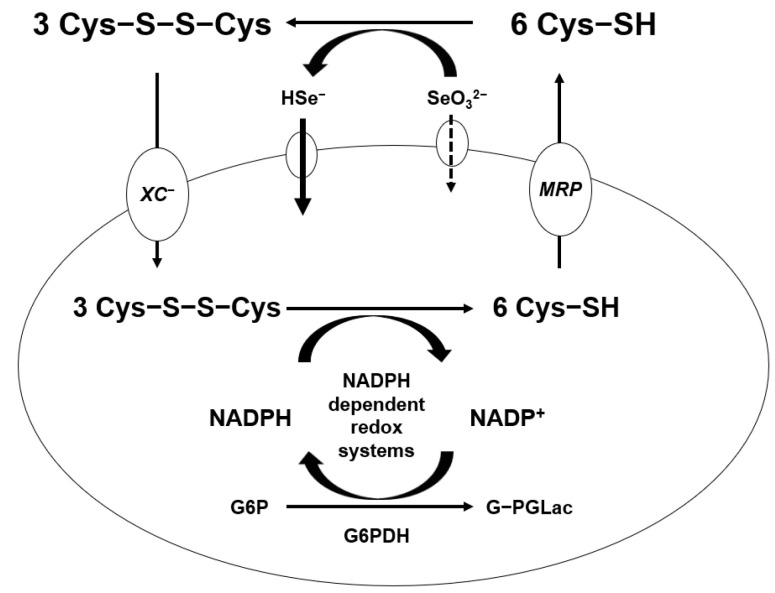
Absorption mechanism of high dose of sodium selenite.

**Figure 4 ijms-22-11844-f004:**
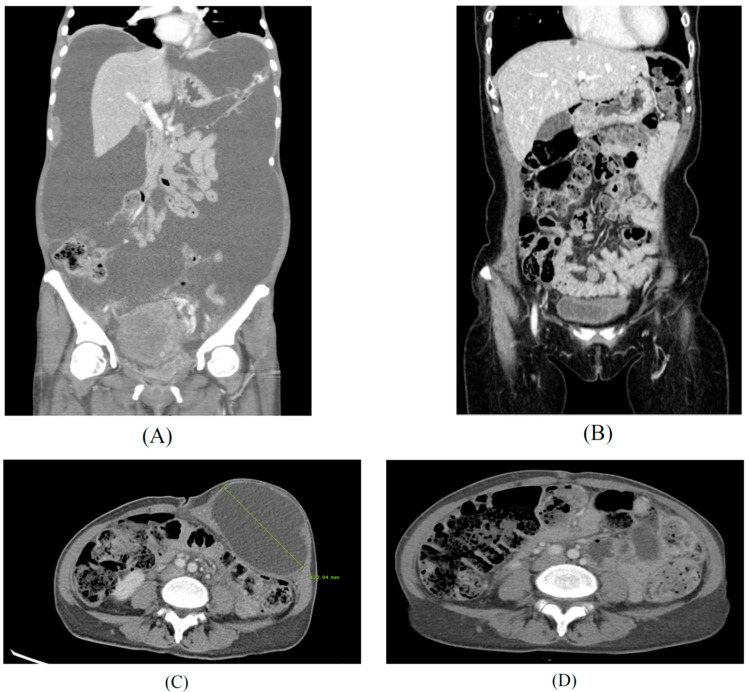
High doses of sodium selenite in treatment of advanced metastasis cancer patients. Case 1: Findings of abdomino-pelvic CT before (**A**) and after (**B**) the multidisciplinary treatment for a 51-year-old patient diagnosed with advanced stage ovarian cancer. Case 2: Findings of abdomino-pelvic CT before (**C**) and after (**D**) the integrative treatment for a 48-year-old patient diagnosed with recurrent ovarian cancer after primary intensive pelvic surgery with adjuvant chemotherapy, since 2017. Peritoneal carcinomatosis with massive ascites (**A**) and no evidence of disease (**B**) after 18 months of chemotherapy, interval debulking surgery, and PARP inhibitor maintenance therapy with high dose sodium selenite treatment according to regimen of Table 1 (SIMCH). Metastatic para-aortic lymph nodes with 12 cm-sized port site metastatic mass at left lower abdominal wall was noted (**C**), and no evidence of disease at left lower abdomen and stable disease of metastatic lymph nodes (**D**) after 16 months of metastatectomy of abdominal wall and maintenance immune checkpoint inhibitor (pembrolizumab) with high dose sodium selenite treatment according to regimen of Table 1 (SIMCH).

**Table 1 ijms-22-11844-t001:** Sodium selenite as integrative medical application to cancer patients at SIMCH.

Indication/Dose		
Pre-treatment after diagnosis	200~300 μg/d	Oral/ample
Intensive care		Ample/injection
Operation	1000~2000 μg/d pre-operation 1~2 h	Ample/injection
Radiation	1000~2000 μg/d pre-radiation 1~2 h [80,81]	Ample/injection
Chemotherapy	2000~3000 μg/d pre-chemotherapy 1~2 h	Ample/injection
Recovery period	500~1000 μg/d	Ample/injection
Multi-organ metastasis	3000~5000 μg/d	Ample/injection
Multidisciplinary treatment failure	5000 μg~10,200 μg/d	Ample/injection
Survivorship clinic	500~1000 μg/d (early)	Ample/injection
	200~300 μg/d (late)	Oral/ample

## Data Availability

The datasets used and/or analyzed in the current study are available from the corresponding author upon reasonable request.

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
