# Peer review of "Antitumor Effects of Selenium"

_ijms, 2021, doi:10.3390/ijms222111844_

Round 1

Reviewer 1 Report

The authors submitted an interesting overview about antitumor effects of selenium. The manuscript is written very well, clearly showing the effects of selenium on cancer cells depending on its concentration and species. Moreover the authors are discussing the clinical application of high doses of SeO32-. Prior to publication I may recommend to consider the following publication: Hans J. Reich and Robert J. Hondal, Why Nature Chose Selenium , ACS Chem. Biol. 2016, 11, 821−841.

Author Response

Reviewer #1 (Reviewer Comments to the Author):
The authors submitted an interesting overview about antitumor effects of selenium. The manuscript is written very well, clearly showing the effects of selenium on cancer cells depending on its concentration and species. Moreover the authors are discussing the clinical application of high doses of SeO32-. Prior to publication I may recommend to consider the following publication: Hans J. Reich and Robert J. Hondal, Why Nature Chose Selenium , ACS Chem. Biol. 2016, 11, 821841.

= Thank you so much for your kind comments. According to your comment, we put the following references in revised manuscript;

“62.Reich, H.J.; Hondal, R.J. Why Nature Chose Selenium. ACS Chem Biol 2016, 11, 821-841, doi:10.1021/acschembio.6b00031.”

Reviewer 2 Report

Dear autors,
The review is very interesting due to the increasing incidence of malignancies. It provides an insight into the additional possibility of treating the disease.  
Text is written according to the instructions. M suggestions are: 1. The abstract is significantly longer than the recommended 200 words.
2. The reference should be standardized and written according to the instructions given in the instructions for the aurora.
3. Emphasize conclusions, recommendations and possible guidelines for further research and possible therapy.

Best regards

Author Response

Reviewer #2 (Reviewer Comments to the Author):
The review is very interesting due to the increasing incidence of malignancies. It provides an insight into the additional possibility of treating the disease. Text is written according to the instructions. M suggestions are:

  1. The abstract is significantly longer than the recommended 200 words.

= Thank you so much for your kind comments. In this revised manuscript, we corrected this as follows;

“Abstracts: Selenium is a ubiquitous element similar to sulfur in its chemical properties. Functions of selenium are diverse as antioxidant, anti-inflammation, increased immunity, reduced cancer incidence, blocking tumor invasion and metastasis, and further clinical application as treatment with radiation and chemotherapy. These functions of selenium are mostly related to oxidation and reduction mechanisms of selenium metabolites. Hydrogen selenide from selenite, and methylselenol (MSeH) from Se-methylselenocyteine (MSeC) and methylseleninicacid(MSeA) are most reactive metabolites produced reactive oxygen species (ROS) and further these metabolites may involve in oxidizing sulfhydryl groups including glutathione. Selenite also reacted with glutathione produces hydrogen selenide via selenodiglutathione (SeDG), which induces cytotoxicity and apoptosis of cancer cells by production of ROS. However, a more pronounced effect was shown in the subsequent treatment of sodium selenite with chemotherapy and radiation therapy. High doses of sodium selenite were effective to increase radiation therapy and to reduce radiation side effects.In our study, advanced cancer patients can tolerate until 5,000 μg of sodium selenite in combination with radiation and chemotherapy since the half-life of sodium selenite may be relatively short and further Se may accumulates more in cancer cells than that of normal cells, which may be toxic to the cancer cells.”

  1. The reference should be standardized and written according to the instructions given in the instructions for the aurora.

= In this revised manuscript, we corrected it all as follows;

“62.Reich, H.J.; Hondal, R.J. Why Nature Chose Selenium. ACS Chem Biol 2016, 11, 821-841, doi:10.1021/acschembio.6b00031.”

  1. Emphasize conclusions, recommendations and possible guidelines for further research and possible therapy.

= In this revised manuscript, we added the following sentence according to your kind comment as follows;

“In conclusion, high dose selenium therapy can be tolerable agent without any significant severe adverse events as an integrative medical treatment combination with chemotherapy, radiotherapy and radical surgery in the advanced stage cancer of patients in chemo resistant status. In the forward, individual take of modification in therapy of dose of duration of treatment with or without combination of standard treatment such as surgery, irradiation, chemotherapy and also immunotherapy. In the future, randomized controlled prospective clinical trials in terms of high dose inorganic selenium treatment will be necessary in the various institutions internationally.”
